# Indoor Radon Levels in Homes and Schools in the Western Cape, South Africa—Results from a Schools Science Outreach Initiative and Corresponding Model Predictions

**DOI:** 10.3390/ijerph20021350

**Published:** 2023-01-11

**Authors:** Abbey Matimba Maheso, Jacques Bezuidenhout, Richard Thomas Newman

**Affiliations:** 1Department of Physics, Stellenbosch University, Stellenbosch 7600, South Africa; 2Faculty of Military Science, Stellenbosch University, Saldanha 7394, South Africa

**Keywords:** indoor radon activity concentration, electret ion chamber, school outreach

## Abstract

We describe a school science outreach initiative that introduced learners to applied nuclear physics research by means of a two-day workshop that involved learners and teachers from 5 schools in the Western Cape province of South Africa. During this workshop, the participants were introduced to the naturally occurring, inert, colorless, and tasteless radioactive gas radon (^222^Rn). During the first day of the workshop, the participants were informed about the detrimental health impacts of inhaling radon and its daughter radionuclides and were shown how indoor radon activity concentrations can be measured using the electret ion chamber (EIC) technology. The learners were then each supplied with a short-term electret (E-PERM, Radelec, Frederick, MD, USA) and associated ion chamber to enable them to make radon measurements in their homes. The teachers in turn were supplied with EICs to enable them make radon measurements in their schools. The participants returned the EICs on the second day of the workshop, one week later. Here, the drop in the potential difference across each electret was measured in order to calculate the average indoor radon activity concentration. A total of 49 indoor radon concentrations were measured. The average indoor radon concentrations were 36 ± 26 Bqm^−3^ in homes and 41 ± 36 Bqm^−3^ in schools, while the highest concentration was found to be 144 Bqm^−3^. These levels were compared to predictions from a model that uses input information about the uranium content associated with the surface geology at each measurement location. The predictions compared well with the measured values.

## 1. Introduction

Radon (^222^Rn) is a naturally occurring radioactive gas directly produced from the decay of radium (^226^Ra) in the decay series of uranium (^238^U). The contribution of radon gas in the air depends on geological parameters such as soils and rocks containing uranium [1]. The outdoor radon concentrations are lower in Bqm^−3^ compared to the higher concentrations indoors due to the additional radon the comes from building materials [2].

According to the U.S. Environmental Protection Agency (US EPA), radon is the second cause of lung cancer deaths after smoking in the United States [3]. The cancer risk in areas with an average radon concentration of 148 Bqm^−3^ is almost 1 in 100 [4]. The United Nations Scientific Committee on the Effects of Atomic Radiation (UNSCEAR) reported that the estimated average per capita dose from radiation sources was 2.4 mSv year^−1^, and 47% is attributed to radon [5]. The US EPA recommended that 148 Bqm^−3^ be the maximum limit of indoor radon concentrations [6]. The EU environmental legislation required that the maximum indoor radon concentration not be higher than 300 Bqm^−3^ [7]. The National Nuclear Regulator (NNR) in South Africa has also set a reference level of 300 Bqm^−3^.

In this study, we compare the measured indoor radon and predicted indoor radon concentrations. A further aim of the study is to develop strategies to access radon concentrations in homes and schools. A big part of this study was conducted in the Cape Flats and Stellenbosch areas located in the Western Cape Province, South Africa, as shown in Figure 1. The underlying geology of the Cape Flats and the surrounding areas comprises the Cape Granite Suite, Kalahari Sand, Malmesbury Shale, and Table Mountain Sandstone, as indicated in Figure 1 (Council for Geoscience, downloaded from https://maps.www.geoscience.org.za) accessed on 1 August 2020.

## 2. Materials and Methods

### 2.1. Study Design

There is no national indoor radon survey program in South Africa. In preparation for a national indoor radon survey, a design of a nationwide radon survey is necessary. Access to homes and schools was achieved through the science school outreach initiative. The outreach activity was carried out following the protocols approved by the Research Ethics Boards in Stellenbosch University and the Western Cape Department of Education (WCED). The guidelines and regulations for a research study involving science learners as participants were carefully followed.

### 2.2. Study Area

Ten schools were initially selected in the Cape Flats and Stellenbosch areas in the Western Cape, South Africa. The locations of the schools were determined using the geographic information system software (qGIS 2.18). The selection was based on the geological characteristics and the population of that area. The geology information was obtained from the 1:200,000-scale national maps developed by the Council for Geoscience in South Africa. The data about the population were obtained from Statistics South Africa (Stats SA). Invitations for participation were sent to schools, and five school learners and one school teacher were identified by the school principal from each school. The school learners’ inclusion criteria included (1) being in the science stream between grades 10 and 11 and (2) residing in the vicinity of the school (not more than 5 km away from school). Only five schools participated in this study, as shown in Figure 2. 

### 2.3. Organization of the Outreach

The school-based outreach was split into two contact sessions and took the form of a workshop. The school learners were engaged in both sessions. In the first session, the learners were engaged through signing the registration and then introduced to the research topic on indoor radon. The learners were then engaged in receiving a clear demonstration from the researchers on how to deploy the radon detectors indoors. At the end of the first session, the radon detectors were distributed to the learners and school teachers along with the questionnaires.

The second session was held a week later. During this session, the learners and school teachers returned the radon detectors and the questionnaires. The learners were engaged in measuring the final voltages on the detectors and calculating the radon concentrations. Then, they were grouped in smaller groups, and each group presented the outcome of the results. 

### 2.4. Questionnaire

Each participant received a questionnaire and instructions on how to use a radon detector. The questionnaires required information about the building materials, house age, floors in the buildings, and floor type. The participants took photographs of the location and the radon detector before and after deployment.

### 2.5. Radon Measurements

Each learner participant received one E-PERM detector purchased from Rad Elec Inc., designed for short-term radon measurements. Each teacher participant received three E-PERMs. In homes, the learners deployed an E-PERM either in the living room or bedroom, as in Figure 3a. In schools, a teacher deployed all three E-PERMs in classrooms and staff rooms, as in Figure 3b. The exposure period for the radon detectors was about one week (3–7 days). 

Since E-PERMs are sensitive to gamma radiation, radon concentration measurements require corrections for cosmic and terrestrial radiation backgrounds [8]. For all E-PERMs, a correction for background gamma radiation of 12 Bqm^−3^ was assumed. The assumption of the background gamma ray value was based on the fact that any value above 12 Bqm^−3^ yields negative values for an indoor radon concentration. The background gamma ray value was set such that the all final values of indoor radon concentrations were ≥0 Bqm^−3^.

### 2.6. Analysis

At the end of the measurement, all E-PERMs were measured for radon concentrations using Equation (1) below:R_n_ = (V_i_ − V_f_/C_F_ × T_d_) − B_γ_(1)
where R_n_ is the mean radon concentration in Bqm^−3^; V_i_ and V_f_ are the voltages on electrets before and after in volts, respectively; C_F_ is the calibration factor for short-term electrets in volts Bqm^−3^ d^−1^; T_d_ is the exposure in days; and B_γ_ is the gamma background correction assumed to be 12.0 Bqm^−3^ [8]. 

Each short-term electret measurement has three contributing sources of measurement uncertainty. E_1_ is the uncertainty associated with the electret thickness and chamber volume, which has been experimentally measured to 5%. E_2_ is the uncertainty associated with the electret voltage reading. E_3_ is the uncertainty associated with the natural gamma ray radiation with an uncertainty of 10%. The total uncertainty, E_tot_ (%), was measured using Equation (2) below.
(2)$$E_{\mathrm{tot}}=\sqrt{E_{1}^{2}+E_{2}^{2}+E_{3}^{2}}$$

### 2.7. Data Processing

The geographic information system software qGIS 2.18 and Microsoft Excel 2010 were used for the data analysis and processing. The Shapiro–Wilk test was applied for data normality testing when comparing measured and predicted indoor radon concentrations.

## 3. Results

The average indoor radon concentrations in homes and schools were found to be 36 ± 26 Bqm^−3^ and 38 ± 36 Bqm^−3^, respectively. The highest indoor radon concentration was measured in a school to be 144 Bqm^−3^. The average indoor radon concentration in old buildings was 39 ± 34 Bqm^−3^, with a highest value of 144 Bqm^−3^. In the new buildings, the average concentration was 24 ± 26 Bqm^−3^, with the highest value of 84 Bqm^−3^. Table 1 indicates that the average indoor radon concentrations measured in the basement and on the ground or first floors were greater than those measured on the second and higher floors. The majority of homes and schools in this study were located on the first or ground floors.

The indoor radon concentrations’ dependence on the building materials used for construction was investigated. Table 1 shows that the average radon concentrations in cement, shacks (made from corrugated iron), and wooden buildings were higher than for brick buildings. However, the highest value of 144 Bqm^−3^ occurred in a brick building. The indoor radon concentrations’ dependence on the flooring type was also investigated. Table 1 indicates that the maximum value for the indoor radon was measured to be 144 Bqm^−3^ in the building with tile flooring. Furthermore, the average indoor radon concentrations in buildings with wooden flooring was relatively higher when compared to other types of flooring.

Figure 4 shows a histogram of the measured indoor radon concentrations in homes and schools. The measured indoor radon concentrations follow a log-normal distribution (dashed line) as expected. The log-normal distribution is defined to be a continuous probability distribution whose logarithm is normally distributed [9]. The logarithms of the indoor radon concentrations in Figure 4 were calculated, and the histogram results followed a normal distribution. The skewness decreased when the logarithm of the indoor radon was included, and the distribution shifted to a normal distribution. The Shapiro–Wilk test measures whether a random sample is taken from a population that is normally distributed. The Shapiro–Wilk test rejected the normality, and the lower radon concentrations demonstrate the reason for this deviation from normality.

Thirty-seven measurement locations (80% of the data) had indoor radon concentrations lower than the world average of 48 Bqm^−3^ [10]. Ninety-six percent of the measured indoor radon concentrations in homes and schools were below the World Health Organization (WHO) action level of 100 Bqm^−3^ [11].

## 4. Modeling Indoor Radon Concentrations

Bezuidenhout [12] mapped uranium levels in South Africa by populating geological units with uranium concentrations relating to rock types, groups, and subgroups. The estimated indoor radon concentrations were then extracted from the radon exhalation rates, which were derived from the uranium concentrations in the underlying rock units. An estimated radon map was then constructed for all areas in South Africa, and this map compared well with the experimental measurements. The experimental radon concentrations for this study were then compared to the estimated radon concentrations for Bezuidenhout for the research location.

Bezuidenhout’s model [12] predicted indoor radon concentrations below 100 Bqm^−3^ for 37 out of 39 measurement locations. The general average indoor radon concentration was 40 Bqm^−3^ with a standard deviation of 30 Bqm^−3^, which compared well with the estimated radon levels. All indoor radon concentrations, apart from two houses, were measured where the underlying lithology comprised various types of sand and ocean sediments. This lithology typically contains low concentrations of uranium. The two houses that were not in sandy areas fell within the Cape Granite Suite that forms the bedrock of this part of the Western Cape, as indicated in Figure 5.

The measured indoor radon concentrations from this study can be compared to the previously published data [13] on home indoor radon levels in the town of Paarl in the Western Cape province. A prominent feature of Paarl is its granite-rich outcrops located mainly on or near Paarl Mountain. Lindsay et al. [13] also used short-term E-PERM electret ion chamber technology for these measurements. The results indicated a strong link between the surrounding geology and elevated levels of indoor radon, as presented in Figure 6 (R. Lindsay, personal communication, 2 December 2021). The houses located less than 2 km away from the granite-rich mountainous areas showed higher radon levels compared to those located more than 3 km away. Bezuidenhout’s model predicts (J. Bezuidenhout, personal communication, 2 December 2021) indoor radon concentrations in excess of 100 Bqm^−3^ for homes situated on the slopes of Paarl Mountain. These predictions compare well with the measured values (R. Lindsay, personal communication, 2 December 2021).

## 5. Conclusions

Indoor radon concentrations were measured in 5 schools and 35 homes in the Cape Flats and Stellenbosch areas in the Western Cape province of South Africa. Access to schools and homes was facilitated by means of a school science outreach initiative that introduced learners and teachers to applied nuclear physics research with a focus on radon in the air and its measurement. The electret ion chamber (E-PERM) technology worked well for this campaign. The results showed that the average indoor radon concentration was below the NNR reference level of 300 Bqm^−3^. No measured radon levels exceeded the USA (EPA) and EU radon limits of 148 Bqm^−3^ and 300 Bqm^−3^, respectively. Only 4% of the indoor radon concentrations were over the WHO reference level of 100 Bqm^−3^. The results obtained in this study will contribute to the development of an indoor radon map for South Africa. This study has shown that it is viable to use a school science outreach initiative as part of a strategy to augment indoor radon level data.

A model developed [12] to predict indoor radon levels by using as an input the geological properties of the measurement locations was found to work well. Thirty-seven out of the 39 measured radon concentrations were consistent with the predicted range (0 to 100 Bqm^−3^) of indoor radon levels. Two measured values were lower than predicted. This may have been due to differences between true and assumed radon exhalation rates at these two locations. This also may have been due to the occupants’ ventilation practices. The model used in this study can further be used to predict which areas in South Africa are radon priority areas (RPAs). This can aid in prioritizing measurements as part of a national indoor radon survey in South Africa.

## Figures and Tables

**Figure 1 ijerph-20-01350-f001:**
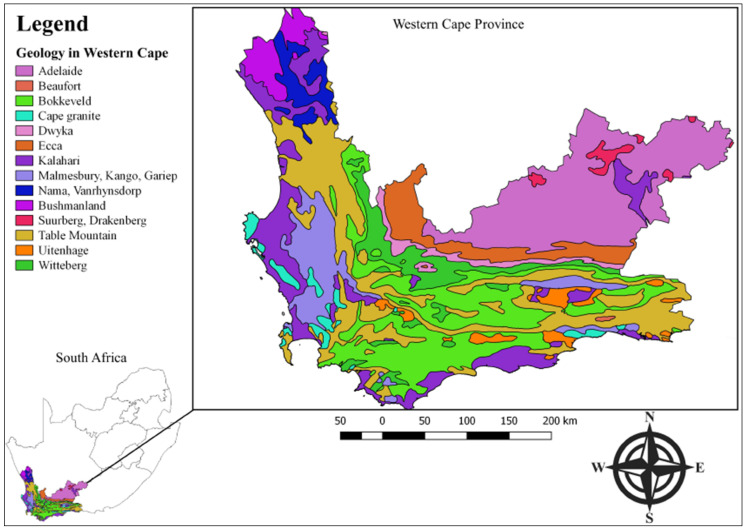
Map of South Africa showing the Western Cape province’s geology.

**Figure 2 ijerph-20-01350-f002:**
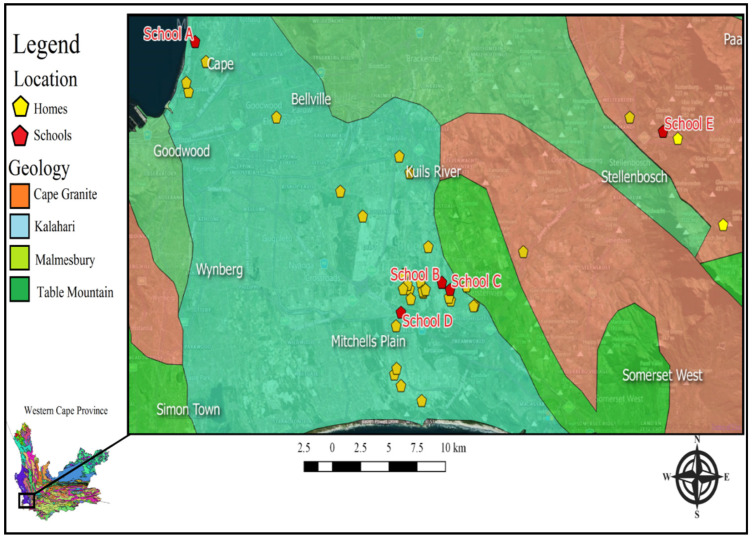
Cape Flats map showing measurement locations of schools and homes and the surrounding geology.

**Figure 3 ijerph-20-01350-f003:**
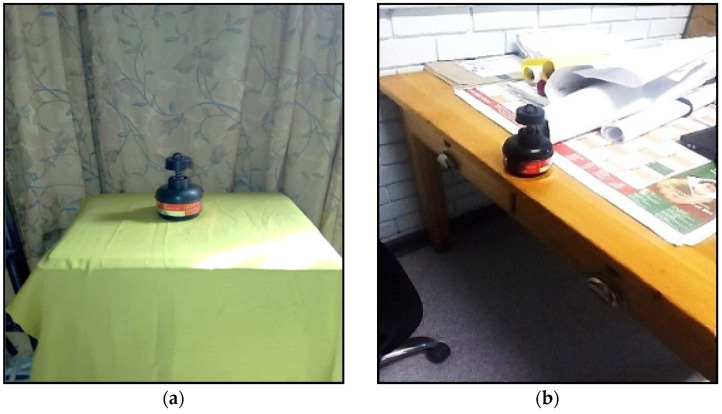
(**a**) Photograph of a deployed E-PERM in one of the learner’s homes and (**b**) in a school staff room.

**Figure 4 ijerph-20-01350-f004:**
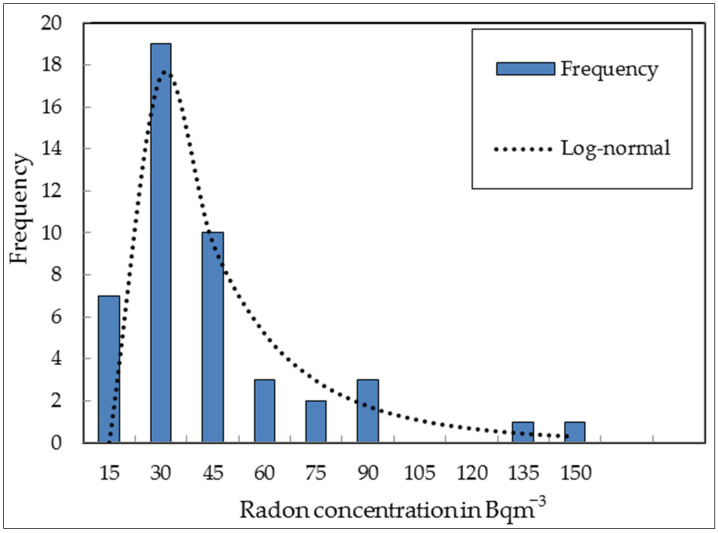
Histogram showing the results of the indoor radon measurements.

**Figure 5 ijerph-20-01350-f005:**
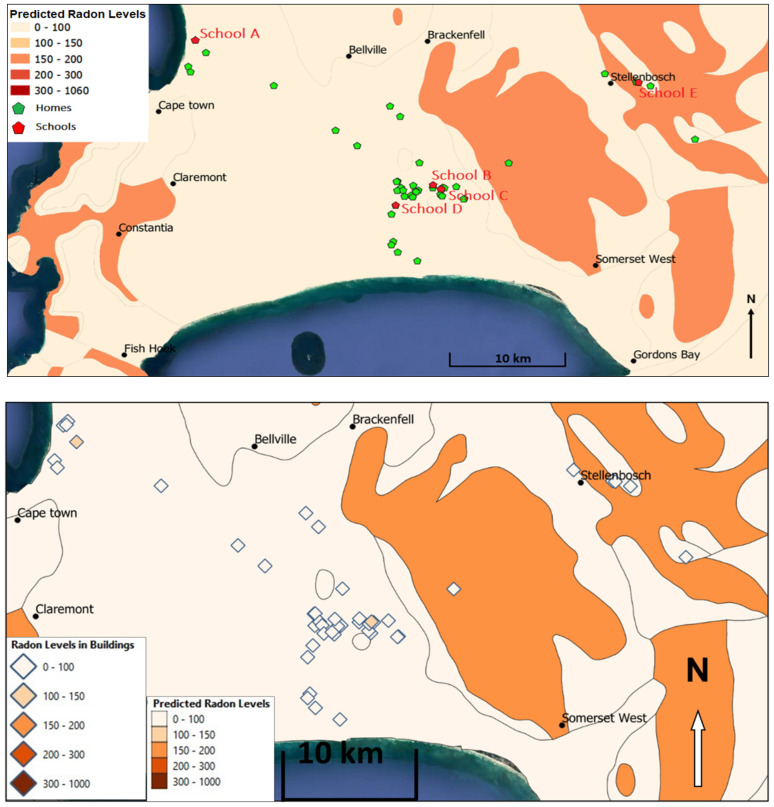
Western Cape map showing the measurement location and measured and predicted indoor radon levels using Bezuidenhout’s model [12].

**Figure 6 ijerph-20-01350-f006:**
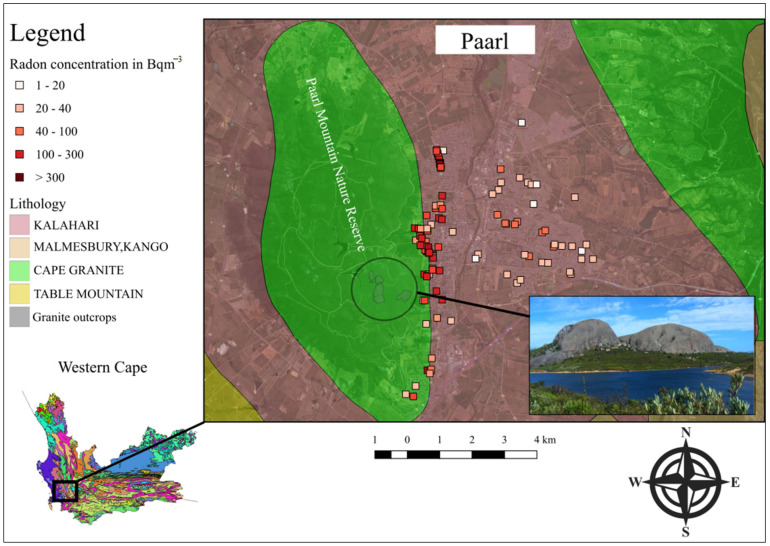
Indoor radon concentrations and the surrounding geology of the Paarl area.

**Table 1 ijerph-20-01350-t001:** Summary of the statistics for measured indoor radon levels (Bqm^−3^).

Variables		N	Mean (SD)	Maximum
Building type	Homes ^1^	35	36 ± 26	127
	Schools ^2^	14	38 ± 36	144
Age of building	Old (>20 years)	22	39 ± 34	144
	Newly built (<20 years)	21	24 ± 26	85
	Not stated	6	37 ± 16	70
Floors in the building	Basement	1	31 ± 2	-
	Ground floor	46	37 ± 29	144
	2nd Floor and up	2	25 ± 16	37
Building materials	Brick	43	36 ± 29	144
	Shacks	4	37 ± 19	60
	Others ^3^	2	52 ± 51	88
Flooring type	Carpets	4	41 ± 30	89
	Concrete	7	27 ± 16	60
	PVC ^4^	5	36 ± 29	82
	Tiles	27	37 ± 32	144
	Wood	6	45 ± 26	88
Overall statistics		49	37 ± 29	144

^1^ Including living rooms and bedrooms. ^2^ Including classrooms and staff rooms. ^3^ Wooden and concrete. ^4^ Polymer of vinyl chloride material.

## Data Availability

Not available.

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
