# Peer review of "Indoor Radon Levels in Homes and Schools in the Western Cape, South Africa—Results from a Schools Science Outreach Initiative and Corresponding Model Predictions"

_ijerph, 2023, doi:10.3390/ijerph20021350_

Round 1

Reviewer 1 Report

The manuscript deals with an important and interesting issue, for the first time in literature, on the  indoor radon concentrations at schools and homes in the Cape Flats and Stellenbosch areas, in the Western Cape Province of South Africa by means of electret ion chamber (E-PERM). Interesting point is that the results obtained in this study will contribute to the development of an indoor radon map for South Africa and a model developed to predict indoor radon levels by using as input the geology.

The manuscript is very well conceived. The manuscript is a significant novel contribution. Summing up, the manuscript itself is well organized and well written in adequate English language. The title is appropriate. The main goal of this work is clear and the activities done to fulfill the goal is described. The analysis on the data is very clear and complete. The main output is clearly described and presented in a good form. The tables and figures and captions are really clear and understandable. The abstract and conclusion fully hit the target and the results obtained in the work, and reflects the overall idea of the manuscript.

The manuscript is fully suitable for the Journal, and perfectly falls into the aims of the Journal. The topic is of interest to readers of the Journal. In conclusion, I recommend the manuscript to be accepted for publication only after several modifications below:

(i) Figure 2 is not cited in the text, please modify and cite it.  

(ii) in the section , when introduce the device E-PERM, please add the following important reference: https://doi.org/10.3390/su12208374

(iii) put caption of Fig. 2 at the bottom of the figure, not in the top:

(iv) put the title of the section near the section, e.g. section 4 and 3 in the new pages. 

(v) please add the information: Funding; Institutional Review Board Statement; Informed Consent Statement; Data Availability Statement; Conflicts of Interest. If not available, write in this way.

(vi) please modify reference 12 and 13, add more information of the conference or the event of the communication etc.

Reviewer 2 Report

The article describes the measurements of radon concentrations in homes and schools carried out by students and teachers. It is a good idea to educate new generations about the risks of NORM. However, we cannot find anything new in this article, but this type of data is useful for determining the global radon dose for a population if participants follow a well-established measurement protocol and follow QA/QC rules

I would like to point out a few issues that need to be clarified before publication

1.  What is the effect of thoron on the measurement?

2.  Have all detectors been placed far enough away from walls, floor and ceiling to avoid thoron effects. The detectors should also be placed at an appropriate distance from the window due to high air exchange.

3.  According to the E-PERM manual, the lowest detection limit is about 5-7 Bq/m3 for a 7-day measurement using short-term detectors. Therefore, results below this value should be excluded from the analysis and from the tables and figures.

4. Figures 4 and 5 show the same results. Delete one of them.

5. Line 213, what is "WHO national level"?

6. Lines 220-221. “This could be due to differences between true and assumed radon exhalation rates at these two locations”.
Please elaborate more about possible discrepancy, not only exhalation rate. What kind of parameters are crucial to estimate indoor radon?

7. Line 222: “radon-prone”. According to the European Atlas of Natural Radiation, Chapter 2, page 51, “the frequently used term 'radon prone area' has been criticised as suggesting that in an area not labelled RPA (radon priority area), no radon problem exists and no action would be required. Due to the high variability of radon, also non-RPA can have high radon concentrations, but with lower frequency, and therefore lower priority might be assigned”.
Therefore, I suggest using “radon priority area (RPA)” instead of “prone area”.

Round 2

Reviewer 2 Report

The authors responded to comments at a moderate level.

Some points still need improvement

Point 5 - "WHO country level".The term "national level" should be replaced.In the WHO Handbook we can find: "WHO proposes a reference level of 100 Bq/m3 to minimize health risks from indoor radon exposure...".Please keep the terms consistent.

Point 6 - The authors did not respond to my comment.Why?There are many papers describing the problem of radon transport and the relationship between gaseous soil radon and indoor radon.

Point 3 - commentary to the authors' answer - there is a misconception of the error (uncertainty) and detection limit. In general, the error is higher for a shorter measurement period and decreases with the measurement time, e.g. see https://doi.org/10.1111/ina.13098 figure 5.
